# Assessment of Hydrocarbon Degradation Potential in Microbial Communities in Arctic Sea Ice

**DOI:** 10.3390/microorganisms10020328

**Published:** 2022-02-01

**Authors:** Angela Peeb, Nga Phuong Dang, Marika Truu, Hiie Nõlvak, Chris Petrich, Jaak Truu

**Affiliations:** 1Institute of Molecular and Cell Biology, University of Tartu, 51010 Tartu, Estonia; marika.truu@ut.ee (M.T.); hiie.nolvak@ut.ee (H.N.); jaak.truu@ut.ee (J.T.); 2Department of Cold Climate Technology, SINTEF Narvik AS, N-8504 Narvik, Norway; nga.dang@sintef.no (N.P.D.); chris.petrich@sintef.no (C.P.)

**Keywords:** shotgun metagenomics, prokaryotic community, Arctic seawater, sea ice, crude oil, hydrocarbon-degrading organisms, hydrocarbon degradation genes

## Abstract

The anthropogenic release of oil hydrocarbons into the cold marine environment is an increasing concern due to the elevated usage of sea routes and the exploration of new oil drilling sites in Arctic areas. The aim of this study was to evaluate prokaryotic community structures and the genetic potential of hydrocarbon degradation in the metagenomes of seawater, sea ice, and crude oil encapsulating the sea ice of the Norwegian fjord, Ofotfjorden. Although the results indicated substantial differences between the structure of prokaryotic communities in seawater and sea ice, the crude oil encapsulating sea ice (SIO) showed increased abundances of many genera-containing hydrocarbon-degrading organisms, including *Bermanella*, *Colwellia*, and *Glaciecola*. Although the metagenome of seawater was rich in a variety of hydrocarbon degradation-related functional genes (HDGs) associated with the metabolism of n-alkanes, and mono- and polyaromatic hydrocarbons, most of the normalized gene counts were highest in the clean sea ice metagenome, whereas in SIO, these counts were the lowest. The long-chain alkane degradation gene *almA* was detected from all the studied metagenomes and its counts exceeded *ladA* and *alkB* counts in both sea ice metagenomes. In addition, *almA* was related to the most diverse group of prokaryotic genera. Almost all 18 good- and high-quality metagenome-assembled genomes (MAGs) had diverse HDGs profiles. The MAGs recovered from the SIO metagenome belonged to the abundant taxa, such as *Glaciecola*, *Bermanella*, and *Rhodobacteracea*, in this environment. The genera associated with HDGs were often previously known as hydrocarbon-degrading genera. However, a substantial number of new associations, either between already known hydrocarbon-degrading genera and new HDGs or between genera not known to contain hydrocarbon degraders and multiple HDGs, were found. The superimposition of the results of comparing HDG associations with taxonomy, the HDG profiles of MAGs, and the full genomes of organisms in the KEGG database suggest that the found relationships need further investigation and verification.

## 1. Introduction

Oceans cover over 70% of the Earth’s surface and the extremely variable conditions in these ecosystems are challenging for microorganisms, such as bacteria or archaea, to adapt to [1]. Crude oil has been part of the marine environment for millions of years, and microbes that use its rich source of energy and carbon are found in seawater, sediments, and shorelines from the tropics to the polar regions [2,3]. However, petroleum hydrocarbons released into the marine environment also derive from anthropogenic activities such as the drilling, manufacturing, storing, and transporting of crude oil and its products [4]. In the Arctic marine environment, the risk of oil spills is increasing due to climate change that enables the prolonged and more frequent usage of Arctic sea routes [5,6].

Marine oil spills are important threats to sea ecosystems, including coastal environments, even more so in colder climates where oil pollution is more persistent due to low temperatures and low light intensity slows the self-dispersing of the spilled oil [7,8]. Oil spill cleanups in Arctic regions also present a challenge due to poorly accessible locations and extreme weather, both of which can complicate or totally impede the usage of traditional oil spill cleanup methods such as booms, skimmers, and pumping systems [9,10]. Therefore, bioremediation techniques, such as harnessing the potential of oil compound degradation by indigenous microbes, have been suggested to be more suitable for such regions due to their relatively easy implementation, cost-effectiveness, and smaller impact on the environment [11,12]. In natural marine environments, the biodegradation of crude oil involves a succession of species in microbial consortia [13]. Microorganisms in the sea, free-living or associated with others, have been studied in laboratories by applying culturing techniques [14,15]. However, an implementation of the whole-genome shotgun sequencing of the total DNA progressively contributes to the understanding of microbial diversity and biochemical pathways [16,17]. Several metagenomic studies have shown microorganisms with the potential to degrade crude oil components, such as alkanes and mono- and polyaromatic hydrocarbons, in Arctic seawater [18,19] and revealed oil-degradation pathways for specific areas by the genes that encoded the enzymes involved in these pathways [20,21,22].

In the case of an oil spill in the Arctic, oil could also become encapsulated in ice, forming oil pockets or leading to the migration of oil through brine channels in the sea ice; both scenarios cause oil to be released back into the water environment sooner or later [23,24]. The microbiome of Arctic sea ice has not yet been sufficiently studied, even though this knowledge would help to develop effective methods for spilled oil cleanup from below or inside of the sea ice [25,26]. Microorganisms from melted sea ice have been shown to degrade oil compounds in the Arctic region, whereas different microbial consortia from seawater have also been reported to take part in this process [5,27,28]. Bacterial genera such as *Pelagibacter*, *Octadecabacter*, *Sulfitobacter*, *Colwellia* [29], *Cycloclasticus* [29,30], *Alcanivorax*, *Marinobacter*, *Thallassolituus* [30], and *Oleispira* [29,30,31], to name a few, are often associated with oil compound degradation in Arctic areas. However, the information about the genetic potential of hydrocarbon degradation in the sea ice of Arctic areas is still not sufficient. Thus, additional studies of microbial responses to spilled oil in ice-covered regions are needed.

The aim of this study was to assess the structure and diversity of the microbial community and estimate its genetic potential for oil hydrocarbon degradation in Arctic seawater, as well as in uncontaminated and crude oil encapsulating sea ice that was formed from this seawater in laboratory conditions.

## 2. Materials and Methods

### 2.1. Sample Collection and Mesocosm Setup

A three-month-long incubation experiment was conducted in Norut’s Cold Climate Laboratory (Norway) to estimate the microbial community’s structure and potential for oil hydrocarbon degradation in uncontaminated Arctic sea ice and in ice with encapsulated Troll B type crude oil (North Sea naphthenic crude oil). Two identical ice growth and incubation sessions were conducted, and for these sessions, the seawater (SW) from the surface of the fjord Ofotfjorden close to the shore area at Kvitvika (68.44208° N 17.38917° E), Norway, was collected with a sterile jug into 40 L containers. For the first session, sampling was performed on 22 October 2015 (SW^1^), and for the second session, sampling was performed on 5 November 2015 (SW^2^). The water temperature was 7.5–8 °C at the sampling site during the sampling period.

In the laboratory, for the ice formation, 120 L mesocosms were set up in pre-cleaned plexiglass tanks insulated at the perimeter and bottom with 5-cm-thick Styrofoam and filled with seawater. Heating elements and a fan were placed underneath the tank to maintain a constant heat flux into the water beneath the ice. Cleaned thermocouples were installed to monitor the temperature in the growing ice and in the water. The lab temperature was set at −1 °C for two days to pre-cool the water in the tank. To initiate the ice growing process, the lab temperature was decreased to −15 °C.

For both sessions, two ice-formation tanks were prepared within 24 h after the water sampling. Tank A was used for growing unpolluted sea ice (SI). In the case of growing ice with encapsulated crude oil (SIO), 250 mL of well-mixed Troll B type crude oil was injected aseptically underneath the ice when the ice thickness reached 7–8 cm in tank B. The sea ice was then allowed to grow about 5 cm thicker under the oil lens. In all cases, the ice was allowed to reach a thickness of about 12–14 cm. Then, from both mesocosms, 9–12.6 kg of the ice was taken for incubation in the freezer at −14 °C for 3 months. After incubation, the ice was melted at room temperature in sterile artificial seawater (Instant Ocean, Blacksburg, VA, USA) at a ratio of 1:1 (*w/w*) for DNA isolation.

### 2.2. DNA Extraction

The SW (8 L) and water obtained from melted ice (9 L from SI, and 10 and 12.5 L from SIO) was aseptically filtered through 0.2 μm Sterivex SVGPL10RC filters (Merck Millipore, Darmstadt, Germany). The Sterivex filters containing DNA were stored at −80 °C until DNA extraction. DNA extraction was conducted using a PowerWater^®^ Sterivex™ DNA isolation kit (MO BIO Laboratories, Inc., Carlsbad, CA, USA), following the instructions of the manufacturer. The quantity and quality of the DNA extracts were determined fluorometrically using the Qubit 2.0 with the Qubit^TM^ dsDNA HS assay kit (Thermo Fisher Scientific Inc., Waltham, MA, USA). The DNA extracts were stored at −20 °C until further analysis.

### 2.3. Metagenome Sequencing

The indexed pair-ended libraries were prepared using a Nextera DNA Sample Preparation Kit (Illumina, San Diego, CA, USA) and a Nextera DNA Sample Preparation Index Kit (Illumina) as described by the manufacturer except for a minor modification; 50 ng of genomic DNA was tagmented at 55 °C for 10 min. The tagmented DNA was amplified with two primers from the Nextera DNA Sample Preparation Index Kit. Each PCR reaction contained 5 μL index 1 primer (N7xx), 5 μL index 2 primer (N5xx), 15 μL NPM (Nextera PCR Master Mix), 5 μL PPC (PCR primer cocktail), and 20 μL tagmented DNA. PCR amplification was carried out with an Eppendorf Mastercycler PCR machine using the following program: 72 °C for 3 min, 98 °C for 30 s, followed by eight cycles of 98 °C for 10 s, 63 °C for 30 s, 72 °C for 3 min, and finally holding at 10 °C. The PCR products were cleaned using Agencourt AMPure XP beads (Beckman Coulter, IN, USA) and the purified PCR products were quantified using a Qubit^TM^ dsDNA HS assay kit (Thermo Fisher Scientific, Inc., Waltham, MA, USA). The sizes of the fragmented libraries were analyzed using a 2100 Bioanalyzer (Agilent Technologies, Inc., Santa Clara, CA, USA). The samples were pooled at a concentration of 4 nM and denatured with 0.2 M NaOH, then diluted to 10 pM with HT1 (hybridization buffer). Samples were sequenced on a MiSeq (Illumina) sequencing platform, using a 2 × 300 cycle V3 kit (Illumina), following the standard Illumina sequencing protocols.

The metagenomic sequence reads are accessible on online repositories. The names of the repositories and accession numbers can be found at: https://www.ebi.ac.uk/ena, PRJEB49528 (accessed on 2 October 2021).

### 2.4. Bioinformatical Analysis

#### 2.4.1. Taxonomic Profiling of the Prokaryotic Community

The quality of the raw metagenomic reads was evaluated using FastQC [32] (v. 0.11.2). Reads shorter than 100 bp and/or containing unambiguous nucleotide(s) were removed, and bases with a quality score less than 20 were trimmed with Cutadapt [33] (v. 1.18). The coverage and diversity metrics of the quality-controlled reads were assessed using Nonpareil [34] (v. 3.3.3). The numbers of reads before and after quality filtering, the counts of classified reads, the metrics for coverage and diversity, and the number of contigs for the metagenomes of each sample are presented in Appendix A. The data of the two metagenomes of the same treatment were pooled together and analyzed further as one metagenome. The composition and structure of the bacterial and archaeal community were classified down to the species level with Kaiju [35] (v. 1.7.3; database: NCBI-nr–30.01.2020). To evaluate the proportion of oil hydrocarbon-degrading organisms (HDO) among prokaryotic genera in the studied metagenomes, the list of genera containing HDOs previously shown to be involved in oil hydrocarbon degradation [36] was updated (Appendix A; *n* = 369) and used as a reference. MEGAHIT [37] (v. 1.2.9) was used to assemble quality-controlled reads into contigs (minimum length of 1000 bp).

#### 2.4.2. Construction of Metagenome-Assembled Genomes

All contigs classified as bacterial or archaeal (Kaiju v. 1.7.3; database: NCBI-nr) were used to assemble metagenome-assembled genomes (MAGs) using MetaBAT2 [38] (default options) (v. 2.15). The quality of MAGs was evaluated using CheckM [39] (v. 1.0.18) and, based on the quality standards from Bowers et al. [40], MAGs with completeness <50% and contamination >10% were excluded from further analysis. The taxonomic classification of good- and high-quality MAGs was performed using three different classifiers, CheckM (v. 1.0.18), Kaiju (v. 1.7.3), and JSpeciesWS [41] (v. 3.5.1), with generally coinciding classification results. However, in some cases, different classifiers classified MAGs at different taxonomical levels. For each MAG, the method giving the lowest taxonomic rank classification was used. A MAG was classified as a pedicular organism with Kaiju when more than 90% of contigs in the MAG were classified as the same organism at the smallest possible rank; if the genus or species rank was possible to identify, the cut-off value for Kaiju identification was 60%. DRep [42] (v. 2.6.2) was used to compare MAGs for their relatedness.

#### 2.4.3. Detection of Hydrocarbon Degradation Genes

Contigs were used to detect 92 genes related to the oil hydrocarbon degradation (HDG) with HMMER [43] (v. 3.3) using KEGG KO [44] HMM (hidden Markov model) profiles (updated 25 December 2019). For the detection of long-chain alkane degradation-related *almA* sequences from contigs, the HMM profile was built with HMMER (v. 3.3) using ClustalW [45] pairwise alignment (default options) [46]. The list of targeted genes along with encoded enzymes is given in Appendix A. If an abbreviation unique to this study was used for the gene, the KEGG code was provided along with the gene abbreviation. Gene counts of each metagenome were normalized using MicrobeCensus [47] (v. 1.1.1) against the reads per kilobase per genome equivalent (RPKG). In the case of HDGs encoding subunits of the same enzyme and forming a gene cluster, the normalized values of the genes encoding structural subunits within the cluster were averaged to be comparable between metagenomes. The gene clusters are marked in the text by the gene symbol that is followed by the letters and numbers (if provided) of the subunits encoding genes. The detected HDGs are considered in the text as four functional groups: (1) genes encoding enzymes involved in the degradation of alkanes; (2) genes encoding enzymes involved in the degradation of monoaromatic compounds (MAHs); (3) genes encoding enzymes involved in the degradation of polyaromatic hydrocarbons (PAHs); and (4) genes encoding enzymes involved in the degradation pathways of various different compounds (defined in the text as various). The pronounced differences in the normalized gene counts of these groups between different environments were determined by calculating the ratios of the normalized gene counts between the metagenomes. The average and standard deviation values of these ratios were calculated for each compound group, and genes staying outside of the range of the ratio’s standard deviation were considered to be substantially different by their counts between the analyzed metagenomes. The annotation of HDGs for MAGs was executed as described for the whole community. In the case of gene clusters, ≥50% of genes encoding enzyme subunits had to be detected to count as present in the MAG. Gene annotation for MAGs was executed as described for the whole community. If a gene had an HMM model for multiple subunits, ≥50% of subunits had to be detected to count as present in the MAG.

Contigs containing any of the specific HDGs (genes detected in almost all prokaryotic organisms (MAGs and genera) of this study that also participate in multiple general metabolic pathways were excluded) were classified using Kaiju (v. 1.7.3; database: NCBI-nr) and the top 100 most abundant prokaryotic genera were analyzed for their association with HDGs. The abundances of HDG hits from the contigs of different metagenomes were normalized against the value of the smallest metagenome (SIO). The KEGG BRITE database was used to confirm the hydrocarbon degradation potential of the genera (unique genera) missing from the list of previously described hydrocarbon-degrading genera [34] and the contigs revealed an association with one or several HDGs in this study. For this purpose, the affiliated HDGs in the fully described metagenomes belonging to the organisms of these genera were searched.

## 3. Results

### 3.1. The Microbial Community Structure in Seawater and Sea Ice

Microbial community analysis resulted in 70.5%, 79.7%, and 75.6% of reads classified as bacterial and archaeal in the pooled metagenomes of SW, SI, and SIO, respectively (Appendix A).

The analysis of bacterial and archaeal reads showed that the prokaryotic community structure of SW differed substantially from the SI and SIO communities, which had higher similarity between each other (Figure 1). Although bacterial sequences dominated in all three metagenomes, their proportion was the lowest in the SW metagenome. At the phylum level, all three metagenomes were dominated by Proteobacteria followed by Bacteroidetes and Actinobacteria. However, the number of phyla that exceeded 1% of all sequences and the proportions of these phyla were different between metagenomes of different origin (Appendix A). The highest number (seven) of these phyla was found in the SW metagenome. The proportion of Proteobacteria was almost doubled due to the increased proportions of all its classes, especially the Alpha- and Gammaproteobacteria in the SI and SIO in particular. However, Alphaproteobacteria was a dominant proteobacterial class in all metagenomes (21.3%, 35.4%, and 33.9% in SW, SI, and SIO, respectively). The proportions of two other dominant bacterial phyla, Bacteroidetes and Actinobacteria, in the SW exceeded the respective proportions in the SI and SIO metagenomes by approximately 50%. The proportion of Cyanobacteria, of which the relative abundance was 2% in the SW metagenome, was below 1% in the SI and SIO metagenomes. The most notable variation between these two metagenomes was in the proportion of Actinobacteria, which was 2.3% higher in the SIO metagenome.

More pronounced differences between the SW and SI metagenomes were revealed at the genus level than were revealed between the two ice metagenomes. The two sea ice metagenomes shared 53 from a total of 76 genera, of which 19 were not found in the SW metagenome (Figure 2). The 20 most abundant genera accounted for 12.4% of prokaryotic reads in the SW metagenome, whereas the respective proportion in the SI metagenome was 20.7% and 21.5% in the SIO metagenome. Ca. *Pelagibacter* was the most abundant genus in all metagenomes, but its proportion was much higher in ice prokaryotic communities (10.9% and 9.7% in SI and SIO, respectively) than in the SW metagenome (4.8%). The proportions of Ca. *Thioglobus* and Ca. *Endolissoclinum* followed a similar pattern, whereas *Glaciecola*, *Colwellia*, *Bermanella*, *Marinomonas*, *Pseudoalteromonas*, *Vibrio*, *Ruegeria*, and *Ascidiaceihabitans* were not amongst the 20 most abundant genera in SW but became abundant in the SI and SIO metagenomes (Appendix A). On the other hand, the proportion of *Synechococcus* was 1.7% in SW, whereas in both ice mesocosms, these proportions remained <0.6%. Out of 20 of the most abundant genera, the SW and SI metagenomes shared 12 genera, whereas in the SI and SIO metagenomes, 17 of the 20 most abundant genera overlapped. The proportion of Ca. *Thioglobus* sequences was lower in the SIO metagenome compared to the SI metagenome, whereas the proportions of ten other genera, namely *Glaciecola*, *Sulfitobacter*, *Bermanella*, Ca. *Actinomarina*, Ca. *Puniceispirillum*, *Ascidiaceihabitans*, *Synechococcus*, *Pseudomonas*, *Pseudoalteromonas*, *Roseovarius*, *Vibrio*, *Alteromonas*, and *Ruegeria* showed higher proportions in the SIO metagenomes compared to SI (Figure 1, Appendix A).

The proportions of archaeal sequences were 3.7- and 3.1-fold lower in the SI and SIO metagenomes compared to the SW metagenome (4.3%) (Figure 1 and Appendix A). At the phylum level, all three metagenomes were dominated by Euryarchaeota (3.9% in SW, 0.84% in SI, and 0.81% in SIO), followed by Thaumarchaeota and Ca. Bathyarchaeota. Overall, 22 archaeal phyla were detected from the SW and SIO metagenomes, and 21 from the SI metagenome. The proportions of sequences of less than 20 archaeal genera (18 in SW, 14 in SI and SIO) exceeded 0.001% of all prokaryotic sequences in the studied metagenomes. *Nitrosopumilus* was the most dominant archaeal genus in all metagenomes (0.051% in SW, 0.087% in SI, and 0.16% in SIO). In the SW metagenome, Ca. *Thalassarchaeum* and *Nitrosarchaeum* followed, whereas in SI and SIO metagenomes, Ca. *Nitrosomarinus* and Ca. *Nitrosopelagicus* were the following archaeal genera (Appendix A).

### 3.2. Proportion of Genera Containing Oil Hydrocarbon-Degrading Organisms in Metagenomes

A total of 350 genera out of 369 genera shown to contain hydrocarbon degraders (HDO) were found across all metagenomes. Although the proportion of the most abundant prokaryotic genera between SW and SI were remarkably different, the difference in proportion of HDO sequences was moderate, at 9.35% in the SI metagenome and 7.11% in the SW metagenome. These metagenomes shared the four most abundant genera-containing HDOs: *Planktomarina*, *Pseudoalteromonas*, *Pseudomonas*, and *Sulfitobacter* (Figure 1B,D,E). In the case of the SW metagenome, the genera-containing HDOs were dominated by the sequences of *Planktomarina* (1.58%), *Pseudomonas* (0.20%), and *Sulfitobacter* (0.19%) (Appendix A). *Planktomarina* and *Sulfitobacter* also dominated in the SI metagenome (1.52% and 0.41%, respectively), where the proportion of *Pseudomonas* (0.26%) was also 1.3-fold higher than in SW. Moreover, *Glaciecola* (0.42%) appeared among the community dominants and the proportions of *Colwellia* (0.37%) and *Bermanella* (0.35%) were 37- and 35-fold higher in the SI metagenome compared to the SW metagenome. The SIO metagenome showcased the highest proportion of HDO sequences (12.16%), which increased by 2.81% and 5.05% compared to the SI and SW metagenomes, respectively. SI and SIO shared the ten most abundant genera-containing HDOs: *Bermanella*, *Colwellia*, *Glaciecola*, *Planktomarina*, *Pseudoalteromonas*, *Pseudomonas*, *Roseovarius*, *Ruegeria*, *Sulfitobacter*, and *Vibrio*, of which all the proportions (except for *Planktomarina*) were higher in the SIO metagenome (Appendix A). The three most dominant HDO-containing genera in the SIO metagenome were *Colwellia* (1.74%), followed by *Planktomarina* (1.11%) and *Glaciecola* (1.05%). Notably, the genera *Paraglaciecola* and *Shewanella* appeared among the dominant section of HDO-containing genera only in the SIO metagenome.

### 3.3. The Abundance of Genes Associated with Hydrocarbon Degradation in Metagenomes

A total of 92 HMM profiles were searched against the analyzed metagenomes. Although the HMM profiles for genes or gene subunits associated with the aerobic degradation of oil compounds were abundant in all metagenomes, genes encoding enzymes for anaerobic pathways were not detected in any of them. The sum of the normalized counts of sequences of all the analyzed hydrocarbon degradation gene groups were highest in the SI metagenome (146 RPKG) followed by SW (125 RPKG) and SIO (118 RPKG) metagenomes. 

In the case of alkane degradation genes, sequences of ten targeted genes were detected, and their counts ranged between 0.001 to 3.81 RPKG across all metagenomes (Appendix A). Although the difference in gene counts between metagenomes stayed in the range of the standard deviation (SD) for alkane degradation genes, the short-chain alkane degradation-related *bmoBCDXYZ* cluster stood out from the group due to its doubled count in SI compared to SW, whereas in SIO, this cluster was not detected (Figure 3A). The oxidoreductase, *r**ubB,* was the most abundant in all metagenomes with 6.8% and 9.7% higher counts detected for SIO compared to SW and SI, respectively. Long-chain alkane degradation-related *almA* and *ladA* followed, being the highest in SI and the lowest in SIO (38% and 41% lower, respectively). The medium-chain alkane degradation-related gene *alkB1_2* showed higher normalized gene counts in the SW metagenome.

Out of 38 tested monoaromatic hydrocarbon (MAH) degradation genes, sequences for 35 genes, including genes from nine gene clusters and six genes encoding oxidoreductases, were detected across all three metagenomes, with normalized counts ranging from 0 to 26.8 RPKG (Appendix A). Across all metagenomes, the most abundant gene clusters were *etbAabc*, *hcaB*, *hcaCDEF*, *xylAM*, and *todABC1C2*. The highest differences between SI and SW were revealed for *phe*, *bsdCD*, and *bsdC*, which were more abundant (by 32%, 49%, and 41%, respectively) in SI, and the lowest for *cymAab* and *dmpKLMNOP*, which were more abundant (by 27% and 33%, respectively) in SW (Figure 3B). The metagenome of SIO had the lowest normalized gene counts of almost all targeted MAH genes, except for *xylAM*, whose proportion exceeded the proportions of this cluster in the SW and SI metagenomes by 6.3% and 10%, respectively.

All 12 targeted polyaromatic hydrocarbon (PAHs) degradation-related genes were detected across the metagenomes. *NahB*, *phdK*, and *gst* had the highest representation in all three metagenomes (Appendix A). An exception was *nidB* from cluster *nidAB* in the SIO metagenome, the sequences of which were not detected from this metagenome. Most of the tested genes of this group were more abundant in the SI metagenome compared to the SW metagenome, with the more pronounced differences in HPGDS and *gst* counts (by 44% and 36%, respectively) (Figure 3C). However, considerably higher abundances of CYP2A6 and CYP3A4 in the SW metagenome compared to the SI metagenome (differences of 24% and 29%, respectively) were also revealed. The counts of these two genes also showed big differences between SI and SIO metagenomes. Although CYP2A6 abundance was higher (by 7%) in the SIO metagenome, the abundance of CYP3A4 and the abundances of *nahC* and *nidAB* in the SI metagenomes exceeded the abundances in the SIO metagenome by 25%, 46%, and 60%, respectively.

In total, 23 genes out of 26 encoding enzymes involved in the degradation of several types of hydrocarbon compounds (various group) were found from the three metagenomes. The normalized counts for the detected genes of this group ranged from 0.10 to 12.7 RPKG and were the highest for *phdE*, followed by *adh*, *adhP*, and ALDH (Appendix A). The abundances of all detected genes in the SI metagenome exceeded the respective abundances in the SW metagenome (Figure 3D). 

### 3.4. Association of Hydrocarbon Degradation Genes with Community Phylogenic Composition

Across all metagenomes, the contigs detected to contain any of the targeted genes specific to oil hydrocarbon degradation were assigned to 285 prokaryotic genera. The genes also participating in multiple general metabolic pathways of prokaryotes (*rubB*, *a-adh*, *ADH1*, *hcaB*, *ped*, *phe*, *gst*, *nahB*, *phdK*, *adh*, *adhE*, *adhP*, *ALDH*, *frmA*, *phdE*, and *sdh*) were excluded from this analysis. The contigs containing HDGs of the SW metagenome were assigned to 148 prokaryotic genera, whereas in SI and SIO metagenomes, this number was 185 and 101, respectively. Thirty-three prokaryotic genera, namely Alphaproteobacterial *Ascidiaceihabitans*, *Bradyrhizobium*, Ca. *Pelagibacter*, Ca. *Puniceispirillum*, *Celeribacter*, *Cohaesibacter*, *Emcibacter*, *Epibacterium*, *Leisingera*, *Lentibacter*, *Maritalea*, *Octadecabacter*, *Pelagicola*, *Planktomarina*, *Planktotalea*, *Pseudooceanicola*, *Rhodobacter*, *Roseobacter*, *Roseovarius*, *Salipiger*, *Sphingomonas* and *Sulfitobacter*, Betaproteobacterial *Variovorax*, Gammaproteobacterial *Amphritea*, Ca. *Thioglobus*, *Halioglobus*, *Methylophaga*, *Pseudomonas*, *Thalassotalea*, and *Umboniibacter*, as well as *Polaribacter* from Bacteroidetes and *Nitrosopumilus* from Thaumarchaeota, contained HDGs from all four functional gene groups analyzed (genes related to alkane, MAHs, PAHs, and various types of hydrocarbon degradation), and were simultaneously amongst the top 100 genera based on the normalized counts in each compound group. A notable number of the HDG and genera associations found were unique to this study (Figure 4, Figure 5, Figure 6 and Figure 7).

#### 3.4.1. Associations of Alkane Degradation Genes

In total, the specific group of alkane degradation genes was associated with 59 prokaryotic genera (Figure 4, Appendix A). Similar numbers of such genera were detected from SI and SW metagenomes (35 and 34, respectively), whereas for the SIO metagenome, this number was substantially lower (15). Only *Ascidiaceihabitans*, Ca. *Pelagibacter*, Ca. *Thioglobus*, *Planktomarina*, and *Sulfitobacter* showed associations with alkane degradation genes in all metagenomes. The contigs containing the long-chain alkane degradation-related *almA* gene were affiliated with the highest number (33) of bacterial genera, 15 of which have not previously been described as HDOs in the literature. *LadA* and *alkB1_2* genes were affiliated with 14 and 13 genera, out of which nine and seven associations, respectively, were unique to this study. Often, the genera were associated with more than one alkane degradation gene and the associations between genes and genera were dissimilar between the metagenomes. 

There were also differences in the normalized abundances of genes related to certain genera between the three metagenomes. For example, sequences of *alkB1_2* associated with *Bermanella* were most abundant in the SIO metagenome. *Bermanella* was also associated with contigs containing *almA* in the SIO metagenome as often as *Bradyrhizobium*; however, the sequences associated with the latter were abundant in the SW and SI metagenomes as well. Although *ladA*-containing contigs were mostly found from the SW and SI metagenomes, they were not very abundant in these metagenomes (Appendix A).

#### 3.4.2. Associations of the Monoaromatic Hydrocarbon Degradation Genes

In the case of MAH degradation genes, 190 prokaryotic genera were associated with genes of this group. Again, the highest number (118) of genera associated with these genes were found from the SI metagenome. In the SW and SIO metagenomes, 97 and 64 such genera were detected, respectively. Within the top 100 most abundant genera-containing MAH degradation genes, approximately 1/5, mostly from Alpha- and Gammaproteobacteria, were shared by all metagenomes, and many of the revealed associations were unique for this study (Figure 5, Appendix A). *XylC*, related to toluene and xylene degradation, was associated with the highest number (135) of different prokaryotic genera among this functional group. A large part (48) of these genera were amongst the 100 most abundant genera over all metagenomes and 32 of them have not been previously associated in the literature with a hydrocarbon degradation function. In all metagenomes, the sequences of *xylC* were most often affiliated as belonging to Ca. *Pelagibacter*, Ca. *Thioglobus*, *Planktomarina*, and *Sulfitobacter*. The toluene degradation-related *pchCF* cluster was associated with 12 different genera, of which *Synechococcus* was abundant in the SW metagenome, *Planktomarina* and Ca. *Pelagibacter* in the SW and SI metagenomes, and Ca. *Thioglobus* in the SI and SIO metagenomes. Several genera were associated with more than one HDG of this group. In all metagenomes, Ca. *Puniceispirillum*, Ca. *Pelagibacter*, and Ca. *Thioglobus* were related to the *etpAabc*, *hcaBCDEF*, and *todABC1C2* gene clusters; *Planktomarina*, in addition to the three latter metagenomes, were also related to *cymA* and *xylAM,* and Ca. *Thioglobus* to the *dmpKLMNOP*, and *tmoABC1C2* gene clusters. In the SW and SI metagenomes, the sequences of *Roseovarius* were affiliated with *etpAabcd*, *hcaBCDEF* and, *todABC1C2*, and in the SI and SIO metagenomes, *Bermanella* and *Oleispira* sequences were affiliated with *cymAa* and *xylAM*.

Two archaeal genera were also associated with the *bsd* genes of this group. The contigs of *Nitrosopumilus* contained sequences of *bsdC2* in the SW and SI metagenomes, and the Ca. *Nitrosomarinus* contained sequences of clustered *bsdC1D* in the SW and SIO metagenomes.

#### 3.4.3. Associations of the Polyaromatic Hydrocarbon Degradation Genes

The contigs containing PAH degradation genes were affiliated with a total of 79 prokaryotic genera. Similar to the other functional gene groups, the highest number (51) of genera was detected for the SI metagenome followed by the SW and then by the SIO metagenome (33 and 20, respectively). All three metagenomes shared approximately 6% of the genera (all from Alpha- and Gammaproteobacteria), and again, many of the genera were not shown to previously contain hydrocarbon degraders (Figure 6, Appendix A). The analysis revealed 30 prokaryotic genera associated with the PAH dehydrogenase encoding *nidAB* cluster, of which 12 were specific to the SW metagenome and 10 to the SI metagenome, whereas the SIO metagenome did not contain specific genera. Ca. *Puniceispirillum*, Ca. *Pelagibacter*, Ca. *Thioglobus*, and *Planktomarina* were shared by all metagenomes, and contigs containing genes of the *nidAB* cluster were most often affiliated with these genera. Some relationships, such as the associations between *Bacillus* and *nidAB*, as well as those between *Ascidiaceihabitans* and 1,2-dihydroxynaphthalene dioxygenase encoding *nahC*, were only revealed in the two sea ice metagenomes.

#### 3.4.4. Associations of the Genes Participating in the Degradation of Various Hydrocarbons

The genes related to the degradation of several types of hydrocarbons (various groups) were associated with a total of 139 prokaryotic genera across all metagenomes. Again, the highest number of genera (88) were found in the SI metagenome. The SW metagenome followed with 69 genera, and only 41 were found in the SIO metagenome. All three metagenomes shared 12 of the top 100 most abundant genera that belonged to Alpha- and Gammaproteobacteria, and again, many of these genera were not previously shown to contain hydrocarbon-degrading organisms (Figure 7, Appendix A). The sequences of *chnB*, a gene related to the degradation of numerous MAHs and cycloalkanes, was affiliated with a total of 50 genera, whereas the abundant sequences were classified to 26 prokaryotic genera, out of which 11 were unique to this study. The *chnB* sequences classified as Ca. *Pelagibacter* were abundant in all metagenomes. The sequences of naphthalene, toluene, xylene, and the other xenobiotic degradation-related gene, *yaiY*, were classified into 35 genera with 17 of these belonging to the top 100 genera. Similar to *chnB*, the contigs containing sequences of *yaiY* affiliated with Ca. *Pelagibacter* and *Planktomarina* were abundant in all metagenomes, whereas *Colwellia* was especially abundant in the SIO metagenome. Other genes and gene clusters related to PAHs, toluene, xylene, benzoate, and naphthalene degradation in this group expressed lower phylogenetic diversity. There were also several associations that were only specific to one metagenome. In particular, the SIO metagenome stood out due to the high number of such specific relationships, including the only archaeal relationship (between *Nitrosopumilus* and the *yaiY* gene) found in this group.

### 3.5. Metagenome-Assembled Genomes

The assembly of quality, trimmed short reads into contigs resulted in 75,931, 84,483, and 51,480 contigs for SW, SI, and SIO metagenomes, respectively (Appendix A). After assembling the contigs, 68 metagenome-assembled genomes (MAGs) with an average completeness of 46.2% and average contamination of 33.8% were obtained (Appendix A). After quality filtering, 18 MAGs were retained (with an average completeness of 72.8% and average contamination of 2.1%) for further analysis (eight from SW, seven from SI, and three from SIO).

#### 3.5.1. Taxonomic Affiliation of MAGs

The relatedness analysis of all good-quality MAG sequences clustered eight MAGs into four pairs by their estimated high similarity (ANIb > 95%) (Figure 8A). Two pairs were formed by the MAGs obtained from the SW and SI metagenomes that were further classified as *Rhodobacteraceae bacterium* SB2 and *Planktomarina temperata* RCA23 (Appendix A). However, half of the obtained MAGs, including a single archaeal MAG, did not show significant similarity with the other MAGs. The only good-quality archaeal MAG obtained for SW out of three archaeal MAGs recovered across the three metagenomes was classified as Ca. *Poseidoniales archaeon*. Excluding the two MAGs that were highly similar in SW and SI, the rest, including all SIO MAGs, were specific to the metagenome origin.

#### 3.5.2. The Genes Related to Hydrocarbon Degradation in Metagenome-Assembled Genomes

All the obtained good- and high-quality MAGs had gene hits for two or more genes belonging to different HDG groups (Figure 8). Across all functional groups, there were 11 genes present in all these MAGs.

The comparison of the profiles of the functional genes in the MAGs shows that despite the MAGs’ origins (SW, SI, or SIO), the more closely related MAGs, phylogenetically speaking, presented similar gene patterns that were often distinctive from the more phylogenetically distant MAGs. For instance, the four MAGs from the family *Rhodobacteracea* (SW7b, SI15b, SI21b, and SIO12b) all had the highest number of different alkane degradation genes among all MAGs. An exception was the MAG affiliated with an organism from the family *Haliaceae* (SW19b) which had a similar set of alkane degradation genes. In addition, almost all targeted genes of PAH degradation and various groups were found in the *Rhodobacteraceal* MAGs, and only some differences between these MAGs were revealed at the species level in terms of MAH degradation (*cymAab*, *xylAM*, *xylM*) and universal gene groups (*pcaGH*). The two MAGs affiliated with *P. temperata* showed a rather similar gene pattern to the *Rhodobacterial* MAGs, but they missed the long-chain alkane degradation-related *almA* gene. A group of MAGs that included two MAGs belonging to *Actinobacteria* (SW5b and SI23b), one to *Bacteroidetes* (SW1b) and SW16b, was characterized by the smallest set of hydrocarbon degradation genes and missed all the main alkane degradation genes. In the single obtained archaeal MAG (SW1a), aside from the genes detected in all MAGs, *alkB1_2* and several genes related to the MAH and PAH degradation were also found.

## 4. Discussion

### 4.1. Prokaryotic Community in Arctic Seawater and Sea Ice

The metagenomic analysis of this study revealed notable differences in terms of the structure of prokaryotic communities and hydrocarbon degradation potential between the Arctic seawater and sea ice, whereas the encapsulated oil led to considerable shifts in microbial community structure and hydrocarbon degradation potential in sea ice.

Proteobacteria, a phylum highly dependent on the abundance of primary producers such as algae in cold seawater [25] and sea ice [25,48,49] was dominant in all three metagenomes. In Ofotfjorden seawater, Alphaproteobacteria was the most dominant Proteobacterial class, in contrast with reports on Canadian Arctic seawater [25] and Svalbard seawater [36] where Gammaproteobacteria dominated in the prokaryotic community. This difference might be caused by the seasonal dynamics of seawater microbial communities [20], as the Ofotfjorden samples were collected in autumn (October–November), as opposed to spring (April–May) in the referred studies. The proportions of Proteobacteria (especially Alpha- and Gammaproteobacteria) in both sea ice microbial communities was approximately two-fold higher than in seawater. On the genus level, the Alphaproteobacterial genus Ca. *Pelagibacter* dominated in all metagenomes. However, the proportion of this genus, shown to thrive in environments with low nutrient concentrations [50], was almost doubled in both sea ice metagenomes compared to the SW metagenome. In contrast with the reports concerning seawater [51], we did not see a remarkable negative effect of crude oil on the proportion of this genus in ice. Moreover, the Cyanobacterial genus *Synechococcus* and Actinobacterial genus Ca. *Actinomarina* were prominent in SW, but their proportions decreased in both sea ice metagenomes, probably due to their sensitivity to lower temperatures [52,53]. The higher proportions of several Gammaproteobacterial sea ice-inhabiting genera, such as *Bermanella*, *Colwellia* and *Glaciecola* [48], which are also known to contain oil hydrocarbon-degrading organisms [27,54], in SIO compared to uncontaminated sea ice, suggests that crude oil encapsulated in ice affects the structure of the prokaryotic community, in contrast to some previous reports [28].

In addition to the whole microbial community and analyses its genetic potential, the recovery of MAGs provides significant insight into microbial evolution and metabolism [55]. MAGs are usually retrieved from more abundant reads of more abundant species in the community [56]; however, the outcome is also affected by many other factors, including the genetic diversity of the microbial community, in addition to the metagenome sequencing depth and coverage [57]. In the current study, 68 MAGs were recovered from 17.4 GB of raw data and 18 of those MAGs were of good or high quality. This outcome resembles the results of the other metagenomic studies [58,59,60]. Several of the recovered good-quality MAGs could be classified down to the genus or species level: two MAGs recovered from SIO were classified as *Bermanella* sp. and *Glaciecola* sp. and were the members of the genera abundant in this environment. Two MAGs recovered from the SW and SIO metagenomes were classified as *Rhodobacteraceae bacterium* SB2, and two from the SI and SW metagenomes were classified as *Planktomarina temperata* RCA23. The latter belonged to the genus which is widespread in cold and temperate marine environments [61] and was among the five most abundant prokaryotic genera in the studied environments.

Similar to the report from the Canadian Arctic [25], the proportion of archaea in the prokaryotic community of Ofotfjorden seawater was below 5% and the archaeal proportion was substantially lower in both sea ice metagenomes compared to SW. However, contrary to Canadian Arctic seawater and sea ice, where Thaumarchaeota dominated in the archaeal communities, Euryarchaeota was the predominant archaeal phylum in Ofotfjorden seawater and sea ice, followed by Thaumarchaeota and Ca. Bathyarchaeota. The Thaumarchaeotal genus *Nitrosopumilus*, found abundantly in Arctic seawater [25], was predominant in all archaeal communities in this study, with its proportion in the SIO community exceeding the SW and SI communities 3-fold and 1.8-fold, respectively. One good-quality archaeal MAG, affiliated with Euryarchaeotal Ca. *Poseidoniales archaeon*, was also retrieved from SW in this study.

### 4.2. Hydrocarbon-Degrading Organisms in the Seawater and Sea Ice Communities

The prokaryotic community of Ofotfjorden seawater used in this study appeared to be rich in the variety of microorganisms previously associated with petroleum hydrocarbon degradation, especially from the classes of Alpha- and Gammaproteobacteria. The proportion of genera known to contain HDOs was 2.2% higher in the SI community compared to SW, whereas in SIO, the proportion of HDOs was the highest and exceeded SI by 2.8% suggesting, contrary to some previous reports [28], a selective pressure of crude oil upon the sea ice prokaryotic community. The dominant part of HDO-containing genera was similar in both sea ice metagenomes, consisting of several cold-adapted genera (i.e., *Colwellia*, *Pseudoalteromonas*, *Pseudomonas*, *Bermanella*, *Sulfitobacter*, and *Glaciecola*) [20,62,63]. However, the proportions of all these genera were higher in the SIO metagenome compared to uncontaminated sea ice. In oil-contaminated sea ice, crude oil might provide the microorganisms that are able to use its components as an additional energy and carbon source with a competitive advantage over the other members of the community [64,65]. Moreover, some genera, such as *Paraglaciecola* and *Shewanella*, with known versatile potential for oil hydrocarbon degradation [36,66], emerged among the dominant HDO-containing genera only in the SIO metagenome. All three MAGs derived from the SIO metagenome were affiliated with taxa containing HDOs (*Bermanella*, *Glaciecola*, and *Rhodobacteraceae*) [67,68].

However, the increase in HDO proportion in the oil-contaminated ice prokaryotic community might not translate to significant hydrocarbon degradation activity [28,69], even in the presence of dispersants [27]. At low temperatures, below −5 °C, oil viscosity is high and sea ice is considered impermeable [28,70]. In these conditions, oil can stay in small brine channels where the biodegradation is restricted due to the small contact area between ice-inhabiting HDOs and substrate as well as high salinity [69]; however, in some cases, it can migrate through ice to underlying water [71]. Furthermore, Arctic sea ice is a nutrient-deficient environment [64] where, similarly to Arctic seawater, the microbial community structure is altered due to oil contamination and has considerable genetic potential in terms of the biodegradation of oil compounds. Gleitz et al. [72] determined that in Antarctic sea ice, the brine chemical composition, including oxygen content, was determined by the chemical properties of the ice-forming seawater and that potentially toxic hyperoxia rather than deficiency of oxygen occurs in these ice compartments. It has also been shown that interactions between the oil and the sea ice alters the oil’s physical and thermodynamic properties and the availability of different oil compounds in the brine [73]. The abundance of the microbial community under nutrient-deficient conditions and the low availability of oil hydrocarbons, accompanied by possible hyperoxia conditions, may remain too low for notable oil biodegradation in sea ice [36,69]. However, oil-degrading bacteria from ice may contribute to the microbial community of seawater and to oil biodegradation once the ice melts [28]. Our results indicate that oil in ice caused the rearrangement of the microbial community, which was reflected in the increased relative abundance of specific oil-degrading microbial taxa and in the reduced hydrocarbon-degradation metabolic potential of the community. This result needs verification with quantitative methods in order to fully understand the metagenomic analysis outcome and to relate microbial community data to the biogeochemical processes of oil in ice environments.

### 4.3. Genetic Potential of Prokaryotic Community to Degrade Crude Oil Compounds in Sea Ice

The comparison of normalized counts of hydrocarbon degradation genes in the metagenomes revealed that high gene counts were always recorded for many genes that, aside from hydrocarbon degradation, also participated in multiple general metabolic pathways critical for cell survival in diverse environments. Most of them encode different dehydrogenases (*phdE*, *nahB*, *phdK*, *ped*, and *hcaB*), including alcohol dehydrogenases (*adh*, *adhE*, *adhP*, *ALDH*, and *frmA*). In addition, the methyl succinate synthase encoding cluster *nmsABC*, salicylate hydroxylase encoding *salDH* (K00480), and glutathione transferase encoding *gst* were among these genes. These genes were also detected from all recovered MAGs. A similar tendency for a number of these genes has been recorded in seawater microcosms with dispersed oil and accompanying uncontaminated seawater controls [20]. The most abundant gene in all metagenomes was the alkane degradation-related rubredoxin NAD^+^reductase encoding *rubB*, which was also found from all good-quality MAGs; however, its abundance was considerably higher (7–10%) in SIO compared to the SW metagenome and, in particular, the SI metagenome, coinciding with previous findings by Ribicic et al., 2018 [20]. Similar to this study, all these abundant genes were also detected from all MAGs recovered from the metagenomes of seawater close to Svalbard [36]. Moreover, the analyzed genes encoding electron transport subunits and ferredoxin reductases yielded substantially higher counts than the genes encoding structural subunits of enzymes. Since the HMM models targeting such genes can also pick the reads of genes homologous to the intended target gene [74], these counts were not taken into account when the abundances of gene clusters were estimated.

Although some of the specific HDGs related to the degradation of alkanes (*almA*), MAH (*xylA*, *xylC*), or PAH (*nidA*) were notably more abundant in all three metagenomes than the other HDGs of the same functional group, a distinctive HDGs profile for each studied metagenome was revealed from the analysis. Although the *xyl* genes have often been used as biomarkers in the evaluation of oil degradation [75], *almA*, a gene related to long-chain alkane degradation, has been largely overlooked in the marine environment so far, even though it has been found in diverse environments [46]. Many studies have focused only on the highly variable middle-chain alkane degradation-related *alkB* and have excluded *almA*, probably partly because of the lack of this gene and the respective HMM model in the KEGG database. However, the studies also analyzing *almA* have shown that the abundance of this gene can be comparable or even higher than *alkB* abundance in various environments, including Arctic seawater [36,76,77]. In this study, aside from *almA*, the abundance of another long-chain alkane degradation gene, *ladA*, also exceeded the abundance of *alkB* in both sea ice environments; however, *alkB* dominated in seawater.

In general, the gene counts indicated that the actual potential to degrade alkanes of different chain lengths, as well as mono- and polyaromatic compounds, was the highest in clean sea ice and lowest in the SIO metagenome. This contradicts the study conducted in the Canadian Arctic where lower proportions of gene reads related to the metabolism of aromatic hydrocarbons were found from sea ice metagenomes compared to seawater metagenomes [25] and suggests different environmental conditions and HDO communities in different Arctic regions. In this study, the abundance of PAH degradation-related dioxygenase encoding *nidAB* was increased in sea ice compared to seawater, but in SIO, this cluster was not detected. An exception was the xylene and toluene degradation-related toluene methyl-monooxygenase encoding cluster *xylAM* and benzaldehyde dehydrogenase encoding *xylC*, the abundances of which were the highest in SIO. Moreover, the abundance of the CYP2A6 encoding cytochrome P450 family 2 subfamily A member 6 was increased in SIO compared to the clean sea ice mainly due to the higher gene counts associated with *Ascidiaceihabitans* and *Planktomarina* in the SIO metagenome.

The knowledge of hydrocarbon-degrading organisms and their oil-degradation-related functional gene profiles in Arctic environments has substantially grown in recent years [20,25,28,36,69,78,79]. Many of the abundant functional genes related to the degradation of alkanes, MAHs, and PAHs were found on contigs affiliated to the genera, including *Planktomarina*, *Colwellia*, *Roseobacter*, and *Marinomaonas*, known to contain HDOs. Some of these genera, for instance, the common marine group *Planktomarina* [61], which was also abundant in all three environments of this study, were related to a high number of genes from all studied gene groups; however, the abundances of these genes were not similar between the metagenomes. There were associations that were specific to the ice. For instance, *Colwellia*, *Glaciecola*, *Bermanella*, and *Sulfitobacter* were associated with multiple different key genes and gene clusters related to the degradation pathways of a variety of compounds in sea ice and, especially, in SIO. This finding was also corroborated by versatile functional gene profiles of MAGs recovered from the SIO metagenome and affiliated with the genera *Bermanella* and *Glaciecola*. The *Bermanella*-affiliated MAG showed a somewhat more diverse HDG profile in this study compared to the MAG recovered from oil-contaminated subsurface seawater [20]; however, this difference might also be caused by different MAG coverage metrics between the two studies.

The substantial number of new associations between the genera and oil hydrocarbon degradation-related functional genes found in this study supports the notion that knowledge of the diversity of hydrocarbon-degrading microorganisms is still vastly increasing [3]. Although many of the genera associated with hydrocarbon degradation genes in this study were already known to contain HDOs, we found possible associations between genera and genes of all studied gene groups that have not shown before. For example, the numerous contigs classified as *Bermanella* and *Oleispira* contained sequences of alkane degradation-related *alkB* and *almA* genes in this study, but previously they have been shown to only possess *alkB* [67,80]. Moreover, several genera, such as *Octadecabacter*, which has previously been associated with *alkB* [81], and *Aureimonas*, *Mycolicibacterium*, and *Rhodoplanes*, which had not previously been associated with oil degradation, were associated with the long-chain alkanes degradation gene *ladA* in this study. This discrepancy can be derived from the species-specific differences between this study and the published data, since different species from the same genus can have different functional gene profiles or technical aspects, such as the length of constructed contigs including sequences of functional genes that affect the number of targeted gene hits and the precision of their taxonomic classification [82].

Besides the high abundance of *almA* in the studied samples, the results suggest that a diverse group of bacteria can be related to this gene. Many of these genera were specific only to the seawater *(Nocardia*, *Marinobacter*, *Sulfitobacter*, etc.), clean sea ice (*Shewanella*, *Pseudomonas*, *Pacillicimonas*, etc.) or SIO (*Planktotalea*, *Bermanella*, *Sarandaracinus*, etc.), and more than half of the associated genera were not among the known HDOs.

Furthermore, contigs affiliated with several genera such as Ca. *Pelagibacter*, *Planktomarina*, Ca. *Thioglobus*, and Ca. *Endolissiclinum* (the former two genera are especially widespread in cold and temperate marine environments [61,62]) showed a high number of associations with oil degradation genes. The comparison of the oil hydrocarbon-related gene profile of *Planktomarina* from both the constructed MAGs as well as contigs and functional gene associations with the gene annotations of the sole representative genome of this genus [61] suggests that the number of oil degradation genes found for this genus from metagenomic data can be overestimated. Furthermore, there were numerous associations between genera, including the ammonia-oxidizing archaeal genus *Nitrosopumilus* from the phylum Thaumarchaeota [83], which had not previously been shown to contain HDOs. This group also contained the most abundant genus in all three marine compartments (i.e., Ca. *Pelagibacter*), which was related to many genes from all gene groups. The predominance of Ca. *Pelagibacter* has been shown to occur at the end of the oil biodegradation period in Arctic seawater and is related to the microbial community’s return to baseline conditions preceding contamination [62]; however, the results of this study suggest that the organisms from this genus might directly contribute to hydrocarbon degradation in Arctic seawater and sea ice.

## 5. Conclusions

The structure of the microbial community and its hydrocarbon degradation potential changed during the formation of ice in Arctic seawater, whereas the encapsulated oil altered the microbial community’s structure and hydrocarbon degradation potential in sea ice. The microbial community’s potential to degrade alkanes of different chain lengths and mono- and polyaromatic compounds were highest in clean sea ice and lowest in sea ice with encapsulated oil. The numerous associations between microbial genera and hydrocarbon degradation genes indicate possible new hydrocarbon degradation abilities and new species within these genera that might participate in the hydrocarbon degradation process in the seawater, sea ice, and oil encapsulating sea ice of the Arctic region. However, these taxonomic and functional associations still need to be verified with additional studies. The study results indicate that the *almA* gene should be considered an additional biomarker for evaluating the biodegradation potential of alkanes in the marine environment.

## Figures and Tables

**Figure 1 microorganisms-10-00328-f001:**
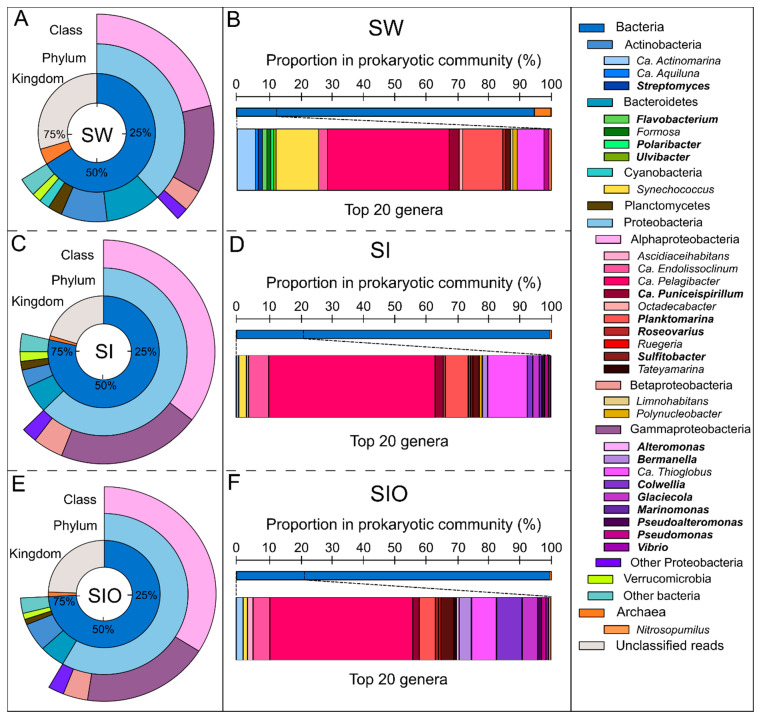
The microbial community structure at kingdom, phylum (>1%), and class (>1%) levels based on all sequences (**A**,**C**,**E**), and the genus (the 20 most abundant prokaryotic genera in each metagenome) level (**B**,**D**,**F**) based on sequences classified as bacterial and archaeal in the metagenomes of seawater (SW), sea ice (SI), and sea ice with encapsulated crude oil (SIO). The genera-containing oil hydrocarbon-degrading organisms are marked in bold.

**Figure 2 microorganisms-10-00328-f002:**
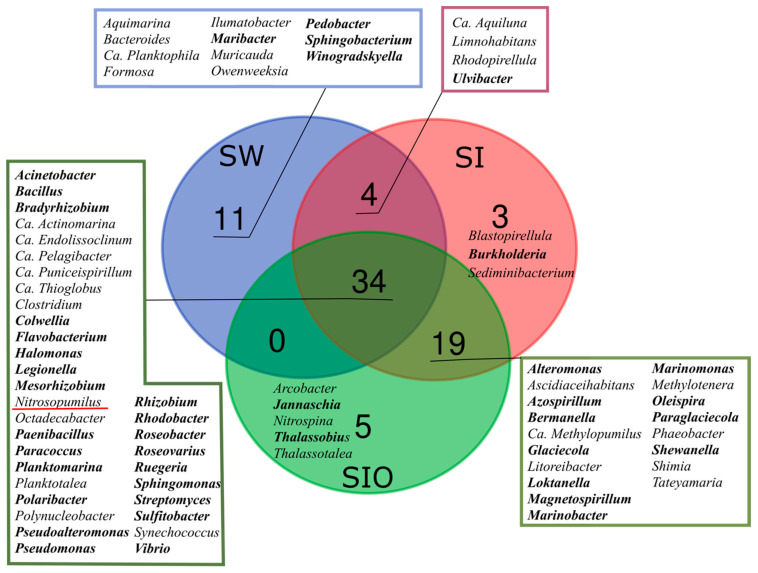
A Venn diagram showing unique and similar prokaryotic genera (>0.05% of prokaryotic community) in seawater (SW), sea ice (SI), and crude oil encapsulating sea ice (SIO) metagenomes. The genera-containing oil hydrocarbon-degrading organisms are marked in bold and archaeal genera are underlined in red.

**Figure 3 microorganisms-10-00328-f003:**
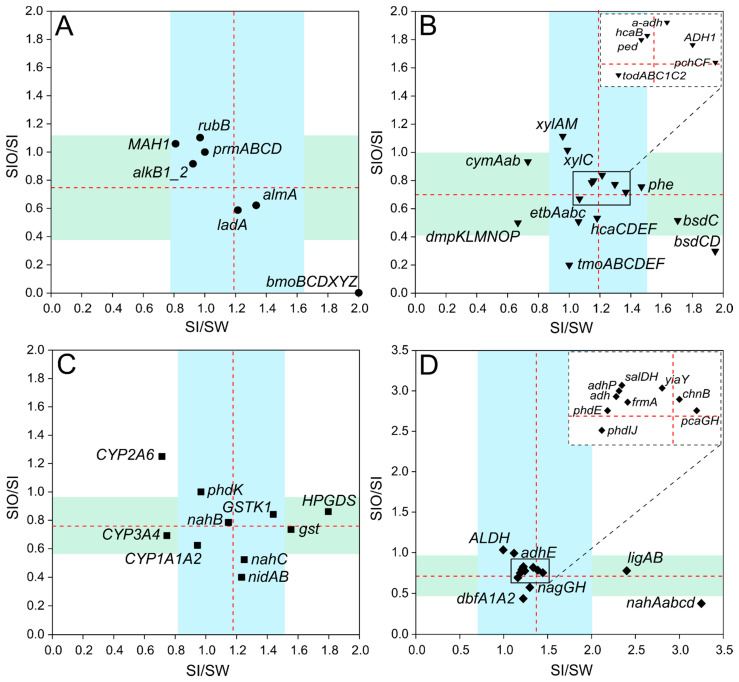
Scatterplots showing ratios of normalized hydrocarbon degradation genes and gene cluster counts between metagenomes of sea ice and seawater (SI/SW) along the *x*-axis, and between sea ice encapsulating crude oil and sea ice (SIO/SI) along the *y*-axis. The ratios are shown on separate subplots for alkane degradation genes (**A**), genes of monocyclic and polycyclic aromatic compounds (**B** and **C**, respectively), and for genes that encode enzymes participating in the degradation pathways of various types of hydrocarbons (**D**). The averages of the ratios of each group are indicated by red dashed lines and standard deviations are indicated by the blue area for the *x*-axis and green area for the *y*-axis. The position of the genes and gene clusters forming tight groups around the crossing of the group averages on the subplots B and D are shown in zoomed windows.

**Figure 4 microorganisms-10-00328-f004:**
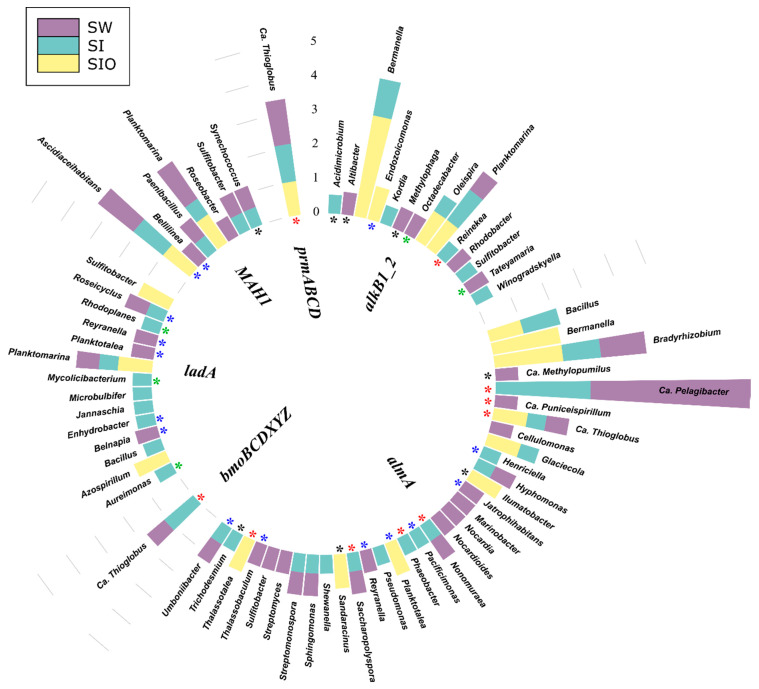
Prokaryotic genera associated with alkane degradation genes and gene clusters in the metagenomes of seawater (SW), sea ice (SI), and sea ice encapsulating crude oil (SIO). Genera without stars correspond to taxa previously shown to involve hydrocarbon degraders. Black stars mark genera that have not been previously shown to contain hydrocarbon degraders in the literature and that the genomes (KEGG BRITE) lack genes annotated to the hydrocarbon degradation genes (HDGs); green stars mark genera that contain organisms with genomes which (KEGG BRITE) have similar hydrocarbon degradation genes (HDGs); red stars mark genera that contain organisms possessing HDGs (KEGG BRITE) different from the current findings; and blue stars mark genera with representatives which were missing from KEGG BRITE.

**Figure 5 microorganisms-10-00328-f005:**
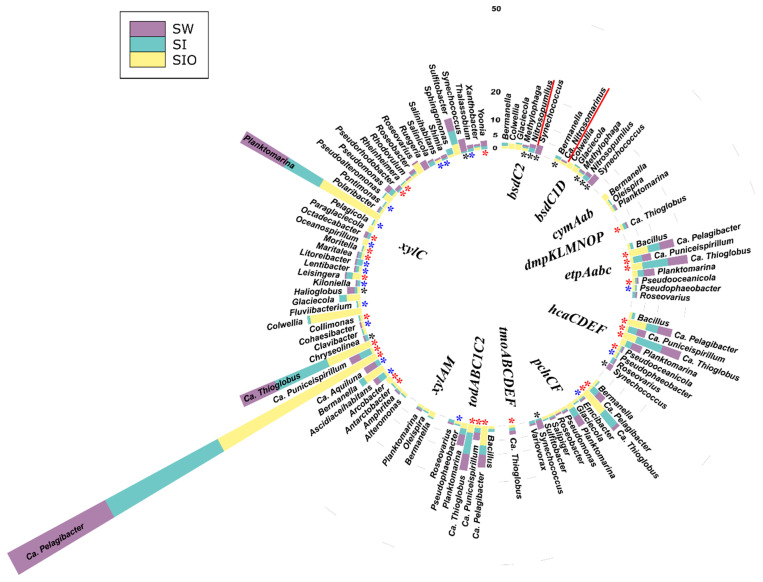
Prokaryotic genera (100 of the most abundant across all metagenomes) associated with monoaromatic hydrocarbon degradation genes and gene clusters in the metagenomes of seawater (SW), sea ice (SI), and sea ice encapsulating crude oil (SIO). Genera without stars correspond to taxa previously shown to involve hydrocarbon degraders. Black stars mark genera that have not been previously shown to contain hydrocarbon degraders in the literature and that the genomes (KEGG BRITE) lack genes annotated to the hydrocarbon degradation genes (HDGs); red stars mark genera that contain organisms possessing HDGs (KEGG BRITE) different from the current findings; and blue stars mark genera with representatives which were missing from KEGG BRITE.

**Figure 6 microorganisms-10-00328-f006:**
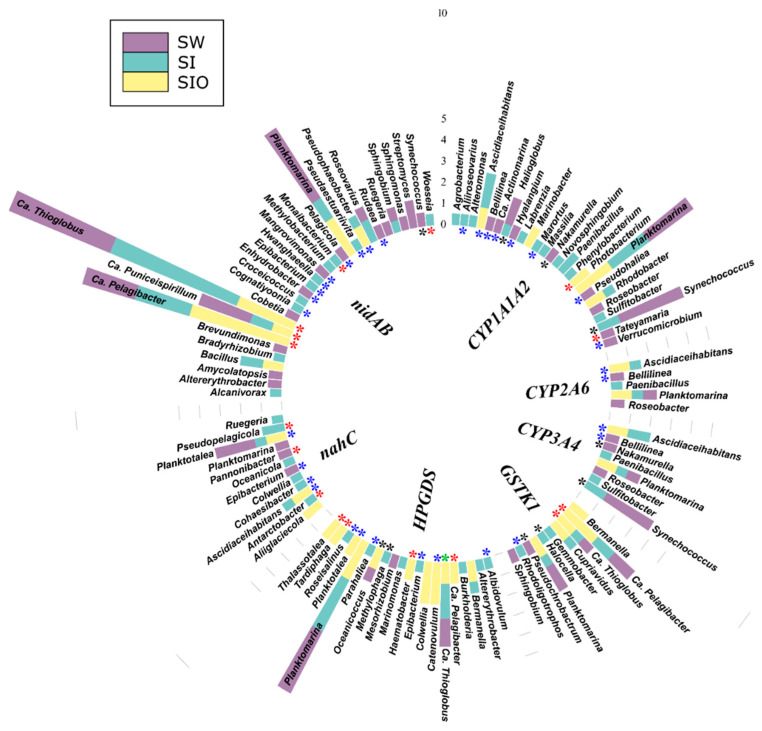
Prokaryotic genera (100 of the most abundant across all metagenomes) associated with polycyclic aromatic hydrocarbon degradation genes and gene clusters in metagenomes of seawater (SW), sea ice (SI), and sea ice encapsulating crude oil (SIO). Genera without stars correspond to taxa previously shown to involve hydrocarbon degraders. Black stars mark genera that have not been previously shown to contain hydrocarbon degraders in the literature and that the genomes (KEGG BRITE) lack genes annotated to the hydrocarbon degradation genes (HDGs); green stars mark genera that contain organisms with genomes which (KEGG BRITE) have similar hydrocarbon degradation genes (HDGs); red stars mark genera that contain organisms possessing HDGs (KEGG BRITE) different from the current findings; and blue stars mark genera with representatives which were missing from KEGG BRITE.

**Figure 7 microorganisms-10-00328-f007:**
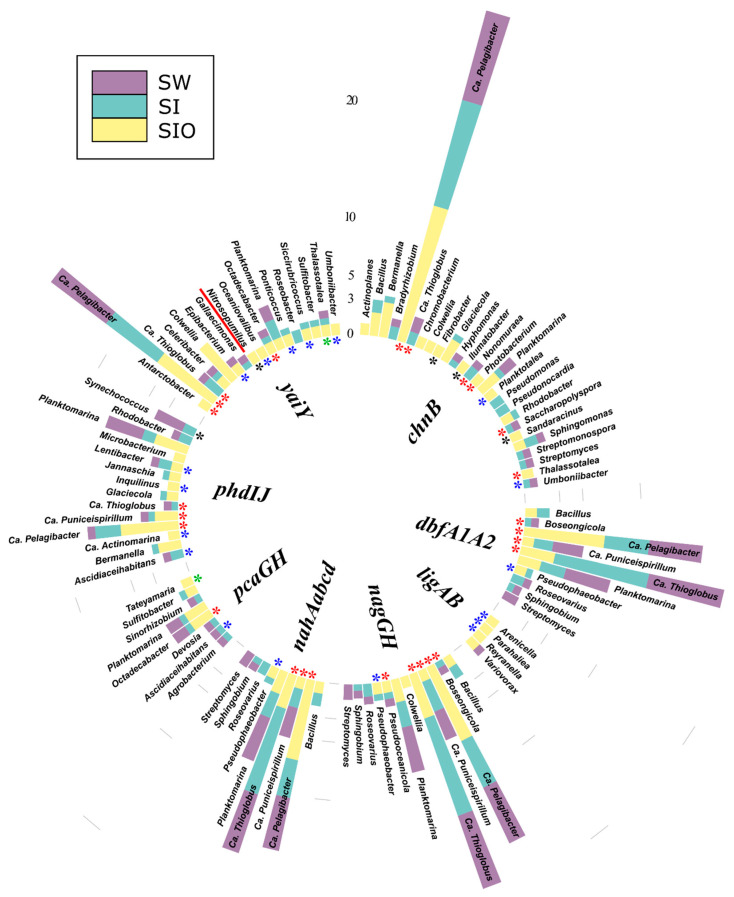
Prokaryotic genera (100 of the most abundant across all metagenomes) associated with the degradation genes of various types of hydrocarbons and gene clusters in the metagenomes of seawater (SW), sea ice (SI), and sea ice encapsulating crude oil (SIO). Genera without stars correspond to taxa previously shown to involve hydrocarbon degraders. Black stars mark genera that have not been previously shown to contain hydrocarbon degraders in the literature and that the genomes (KEGG BRITE) lack genes annotated to the hydrocarbon degradation genes (HDGs); green stars mark genera that contain organisms with genomes which (KEGG BRITE) have similar hydrocarbon degradation genes (HDGs); red stars mark genera that contain organisms possessing HDGs (KEGG BRITE) different from the current findings; and blue stars mark genera with representatives which were missing from KEGG BRITE.

**Figure 8 microorganisms-10-00328-f008:**
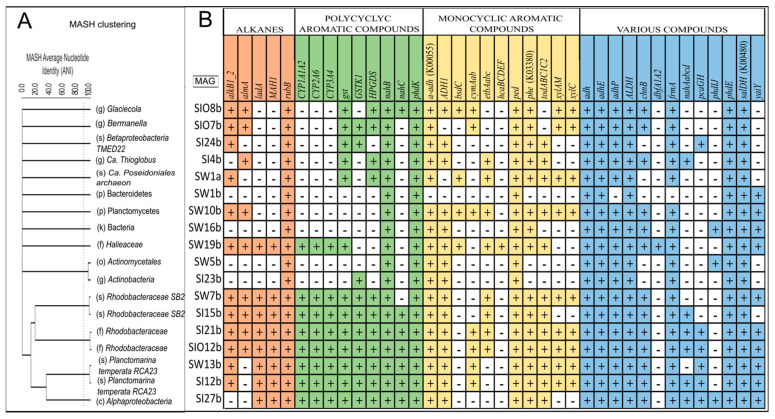
The clustering (relatedness analysis) of MAG sequences (**A**), and the presence of gene clusters encoding enzymes that were involved in the degradation of the crude oil compounds detected in these MAGs (**B**). SW, SI, and SIO in the codes of MAGs refer to seawater, sea ice, and sea ice encapsulating crude oil, respectively, and a and b refer to bacterial or archaeal origin, respectively. The upright dotted line on subplot A distinguishes species-level similarity.

## Data Availability

The data presented in this study are openly available in https://ebi.ac.uk/ena, PRJEB49528 (accessed on 2 October 2021).

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
