# Peer review of "Assessment of Hydrocarbon Degradation Potential in Microbial Communities in Arctic Sea Ice"

_microorganisms, 2022, doi:10.3390/microorganisms10020328_

Round 1
Reviewer 1 Report
The main focus of the paper is DNA sequence handling and comparison of methods to clarify microbial diversity and suggested microbial function in samples. This purpose is fulfilled excellently. The background topic is the microbial community changes observed in differently treated oil spiked arctic waters. The minor change I request is that, in addition to mentioning oil slick (NAPL) as a factor that limits bacterial access to oil and thereby efficiency of oil component degradation, the authors should mention and discuss the slick (and possible other factors) as potentially limiting the oxygen availability to the bacteria in the waters and thereby potentially the community composition. This, however, does not affect the value of the main focus of the paper.
Author Response
Q: “...authors should mention and discuss the slick (and possible other factors) as potentially limiting the oxygen availability to the bacteria in the waters and thereby potentially the community composition.”
A: Discussion text was amended as suggested by the reviewer (section 4.2. page 19)
Reviewer 2 Report
General comments
This study evaluated the hydrocarbon degradation potential of arctic microbial communities in seawater, sea ice and sea ice with encapsulated crude oil using metagenomics. The community was analysed in terms of taxonomy and the hydrocarbon degradation-related functional genes (HDGs) within each metagenome/environment. The data also gave 18
high-quality metagenome-assembled genomes (MAGs), some of which show association of specific genera with hydrocarbon degradation for the first time.
The study is well-written and the figures are of high quality. However, the paper would benefit from reduced length particularly in the Results section. The authors need to reconsider what are the take-home messages of the study and focus on answering/presenting these in a more reader-friendly way. A lot of the information particularly in the Results section does not give valuable information or is too tiring to follow. The authors instead could point only to the interesting aspects but directing the reader to the figures for more detail, which is very well depicted. The authors should bear in mind that they have one metagenome from each environment and some difference in a percentage between one functional gene between Si and Sw, for example, may not have any ecological meaning. In the discussion, the authors should dig deeper into the analysis/explanation of some results and focus on the unique findings of the study i.e., what are the new taxa that are associated with hydrocarbon-degradation based on this study or what is the renewed information about some known HGOs regarding the range of their degradative capabilities. For example, an important genus for HC degradation in cold environments is Oleispira, which so far has been known to degrade short-chain alkanes. The finding that it may also be capable for long-chain alkane degradation is important and should be stressed more along with other such findings. The authors should also focus mainly on the differentially abundant HDOs and MAGs in the oil-contaminated treatment because this is essentially the response of the sea ice microbial community to crude oil contamination. Generally, the data is all there in figures and supplementary material but the text need to be more like a narrative and have a focus on the important take home messages. Essentially, the higher % of a certain gene in one environment is not what matters; we are rather interested in a process i.e., the degradation of a certain HC and what can the genes as a combination tell us about this process in a certain environment.
Minor comments
Abstract
- “In addition, almA was related to the most diverse group of prokaryotic genera” – do you mean that AlmA was present in a diverse group of prokaryotic genera?
Introduction
- However, petroleum hydrocarbons released into the marine environment also derive from anthropogenic activities such as….
- Microorganisms in the sea, free-living or associated with others, have been studied in laboratories applying culturing techniques for a while. – I would remove the phrase “for a while” from this sentence. Culturing is still used and still has merit despite the progress in molecular techniques. Personally, I see the two as kind of complementary.
- Microorganisms from melted sea ice were shown to degrade oil compounds in the Arctic region, whereas communities different from seawater were reported to take part in this process. Please rephrase the bold section. Do you just mean “communities from seawater”?
Materials and Methods
- I am under the impression that all MOBIO products are now Qiagen: https://www.qiagen.com/us/products/discovery-and-translational-research/dna-rna-purification/dna-purification/microbial-dna/dneasy-powerwater-sterivex-kit/
- Section 4.1 “To evaluate the proportion of oil hydrocarbon-degrading organisms (HDO) among prokaryotic genera in the studied metagenomes”
- HDO-containing genera – not sure I understand this, you probably mean genera containing HDOs? Or Taxa containing HDOs?
- In 2.3 Please define HMM
Results
- “The proportions of sequences of less than 20 archaeal genera (18 in SW, 14 in SI and SIO) exceeded 0.001% of all prokaryotic sequences in the studied metagenomes.” This sentence does not make much sense and is probably not necessary since you are mentioning the most abundant archaeal phyla just above. Do you mean the less abundant archaeal phyla or is it indeed genera and you are referring to the 20 most abundant?
- 2 Title: I don’t think there should be a bar between Genera and containing. It’s genera containing oil hydrocarbon-degrading… Please correct throughout the document. You could say HDO-containing genera.
- Section 3.3 – please rephrase the sentence: “The mean of normalized gene count ratios between different metagenomes indicated that the SI metagenome exceeded the other two metagenomes by the total number of gene sequences in each functional group (Figure 3).” You probably mean that the hydrocarbon degradation genes within each functional groups was higher in SI as is the case for the total counts (all functional groups together).
- Section 3.3 is a quite long and difficult to comprehend. The authors whould think of a way to make it more reader friendly. For example, there is no need to mention what is not there (e.g. “Only the genes from cluster nmsABC encoding naphtyl-2-methyl-succinate synthase were not found from any of the metagenomes. Moreover, nagH and nahAd were lacking in the SW metagenome, and dbfA2 and nahAd were not detected from the SIO metagenome”. Also, are there any genes that take part in the degradation of the same compound? Maybe there can be a clustering of this sort in the description with the names of particular genes in parenthesis rather than stating %differences of each gene separately.
- What do you mean by “various types of hydrocarbons”, please specify what is included in this category.
- Figure 4. To be honest, apart from the black stars I don’t understand what kind of information the other stars give us. Can you elaborate. Also, Figure 4 is very informative and understandable so that the text in the corresponding paragraph can be reduced considerably to contain only important information for the main messages stemming from this paper.
- Section 3.5.2, 1st Maybe it’s more useful to tell the reader about the 11 common genes (which are they?) instead of the genes that are NOT present in MAGs.
Discussion
- Proteobacteria are primary producers? Maybe you mean something else here?
- How do the authors explain that the hydrocarbon degradation potential was highest in sea ice and lowest in the SIO? What does this mean for the interpretation of the metagenomic results and their connection to an ecosystem process (i.e., the degradation of hydrocarbons).
Author Response
Q: (General comments). “... the paper would benefit from reduced length particularly in the Results section.”; “... the authors should dig deeper into the analysis/explanation of some results”; “The authors should also focus mainly on the differentially abundant HDOs and MAGs in the oil-contaminated treatment...”
(A1-3)
We reduced the length of Results section text (particularly 3.3 and 3.4). We hesitate to extent discussion more in the case of possible new hydrocarbon degradation related taxa. The reason for this is following. Current study is based on only shotgun metagenome data with limited sequencing depth. In order to eloborate more on possible new HDO taxa or HDO taxa with with more diverse metabolic capablities for oil degradation, we need multi-omic data set and higher sequencing depth. Based on obtained results of this manuscript we performed a new lab experiement with more treatments and replicates, temporal sampling and combination of different molecular microbiology methods. We hope that the results of this new experiment will provide solid proof for findings of this study.
(Minor comments)
(1) Q: ““In addition, almA was related to the most diverse group of prokaryotic genera” – do you mean that AlmA was present in a diverse group of prokaryotic genera?”
A: In that case contigs that were found to contain almA gene were all classified and it resulted in long list of different microorganisms which was more diverse than for other alkane degradation genes.
(2) Q: “However, petroleum hydrocarbons released into the marine environment also derive from anthropogenic activities such as….”
A: Thank you for suggestion, we have included it.
(3) Q: “Microorganisms in the sea, free-living or associated with others, have been studied inlaboratories applying culturing techniques for a while. – I would remove the phrase “for a while” from this sentence. Culturing is still used and still has merit despite the progress in molecular techniques. Personally, I see the two as kind of complementary.”
A: Thank you for suggestion, we have included it.
(4) Q: Microorganisms from melted sea ice were shown to degrade oil compounds in the Arctic region, whereas communities different from seawater were reported to take part in this process. Please rephrase the bold section. Do you just mean “communities from seawater”?
A: This sentence was revised to ‘Microorganisms from melted sea ice were shown to degrade oil compounds in the Arctic region, whereas microbial consortia different from seawater were reported to take part in this process.
(5) Q: “I am under the impression that all MOBIO products are now Qiagen:...”
A: This is correct, Mo Bio was acquired by Qiagen in 2016 and after that the kit names were changed. PowerWater® Sterivex™ DNA isolation kit was Mo Bio brand name and Qiagen used this name for a while.
(6) Q: “Section 4.1 “To evaluate the proportion of oil hydrocarbon-degrading organisms (HDO)among prokaryotic genera in the studied metagenomes””
A: Thank you for suggestion, we have included it.
(7) Q: “HDO-containing genera – not sure I understand this, you probably mean genera containing HDOs? Or Taxa containing HDOs?”
A: You are correct. For better understanding we have rephrased it as suggested.
(8) Q: “In 2.3 Please define HMM”
A: HMM (hidden Markov model) was included in section 2.4.3 where it was first mentioned.
(9) Q: ““The proportions of sequences of less than 20 archaeal genera (18 in SW, 14 in SI and SIO) exceeded 0.001% of all prokaryotic sequences in the studied metagenomes.” This sentence does not make much sense and is probably not necessary since you are mentioning the most abundant archaeal phyla just above. Do you mean the less abundant archaeal phyla or is it indeed genera and you are referring to the 20 most abundant?”
A: The meaning behind the sentence is that we found more than 20 archaeal genera from metagenomes but only 20 of those genera exceeded 0.001% proportion of procaryotic genera. In other words, even if archaea were there their abundance was very low.
(10) Q: 2 Title: I don’t think there should be a bar between Genera and containing. It’s genera containing oil hydrocarbon-degrading… Please correct throughout the document. You could say HDO-containing genera.
A: Thank you, we have included it.
(11) Q: Section 3.3 – please rephrase the sentence: “The mean of normalized gene count ratios between different metagenomes indicated that the SI metagenome exceeded the other two metagenomes by the total number of gene sequences in each functional group (Figure 3).” You probably mean that the hydrocarbon degradation genes within each functional groups was higher in SI as is the case for the total counts (all functional groups together).
A: This sentence means that on average, SI metagenome gene counts were higher than SW and SIO. As this can be depicted from the Fig 3 then we removed this sentence.
(12) Q: Section 3.3 is a quite long and difficult to comprehend. The authors whould think of a way to make it more reader friendly.
A: We shortened this section
(13) Q: What do you mean by “various types of hydrocarbons”, please specify what is included in this category.
A: We have devided genes into 4 groups according to which compound degradation pathway they are included. Some genes, however, are included in multiple compounds degradation pathways, for example gene cluster ligAB, which is associated with degradation of xylene as well as PAHs. Table S3 provides total overview of genes included in various group.
(14) Q: Figure 4. To be honest, apart from the black stars I don’t understand what kind of information the other stars give us. Can you elaborate. Also, Figure 4 is very informative and understandable so that the text in the corresponding paragraph can be reduced considerably to contain only important information for the main messages stemming from this paper.
A: Meaning of stars is elaborated in figure subtexts. Green stars indicate to the genera that have been associated with the same gene before (by KEGG BRITE database) as we did. Red stars indicate to genera that have been associted with some other HDG by KEGG BRITE compared to out findings. Blue stars mark genera that were missing from KEGG BRITE database and we were unable to compare them. We have rephrased subtexts as well.
(15) Q: Section 3.5.2, 1st Maybe it’s more useful to tell the reader about the 11 common genes (which are they?) instead of the genes that are NOT present in MAGs.
A: This sentence was removed from the text.
(16) Q: Proteobacteria are primary producers? Maybe you mean something else here?
A: We have rephrased the sentence for better understanding: “Proteobacteria, a phylum highly dependent of the abundance of primary producers...”.
(17) Q: How do the authors explain that the hydrocarbon degradation potential was highest in sea ice and lowest in the SIO? What does this mean for the interpretation of the metagenomic results and their connection to an ecosystem process (i.e., the degradation of hydrocarbons).
A: This result certainly needs verification with quantitative microbiological methods. The microbial community hydrocarbon degradation potential may be affected by low nutrient and possible hyperoxia conditions in sea-ice brine, plus there could be the effect of altered oil properties and partitioning in ice. We amended Discussion section in case of this matter (4.2 page 19).
Reviewer 3 Report
The submitted manuscript describes a unique experimental approach, where sea ice is generated under laboratory conditions, with or without oil enclosure, incubated at freezing temperatures and then sampled for microbial community analysis through shotgun metagenomics.
The most interesting aspect of the results appears to be the discovery of novel hydrocarbon-degradation gene-containing organisms and identification of novel hydrocarbon-degradation related genes in known hydrocarbon degrading organisms.
Unfortunately the study design didn't allow for replication which excluded the possibility to identify statistically significant changes between the different groups.
Nevertheless, the authors performed an extremely thorough analysis of the obtained metagenomic data, which offers a wealth of highly valuable information to fellow researchers in the field of oil biodegradation.
There is still a need for more knowledge regarding the distribution of hydrocarbon-degrading microorganisms and their genetic potential in both the "phylogenetic space" and also in real geological space.
To that end, this study contributes significantly to the exploration of the Arctic seawater and sea ice microbiome and its potential for natural oil spill response.
Author Response
(1) Q: “Unfortunately the study design didn't allow for replication which excluded the possibility to identify statistically significant changes between the different groups.”
(2) Q: “There is still a need for more knowledge regarding the distribution of hydrocarbon-degrading microorganisms and their genetic potential in both the "phylogenetic space" and also in real geological space. ”
A: We agree with both statments and as for the statistics we take it into count for our future projects.
Reviewer 4 Report
I have received the article entitled: “Assessment of Hydrocarbon Degradation Potential in Microbial Communities in Arctic Sea Ice”, submitted to Microorganisms, for a review.
- Overall, the manuscript (especially in the result section) is a bit too detailed and too long. I would strongly recommend to shorten this part, to not overwhelm the Readers with too much data.
- The explanation of one of the most interesting aspects of this manuscript, namely that the gene counts indicated that the actual potential to hydrocarbon degradation was the highest in clean sea ice and lowest in SIO metagenome was a bit insufficiently discussed. I would expect to obtain opposite results, thus the proper discussion of this issue is missing. Please, try to evaluate this aspect more deeply.
- What is the authors' opinion on the incubation of ice for a longer period of 6-12 months? I think it would be extremely interesting to collate this research.
Specific comments:
- Please indicate precisely what does it mean “shortly” – “For both sessions, two ice-formation tanks were prepared shortly after the water sampling.”
- Figure 1: I would recommend showing data not vertically but horizontally. In my opinion, the figure would be a bit clearer at first glance. However, I leave this decision to the authors.
- “A total of 350 genera out of 368 genera” please indicate the figure/table where the Reads can verify this information.
- Figure 3: Please add red dashed lines also in zoomed windows.
- Is MAH abbreviation explained in the manuscript?
- It would be good to indicate genes related to the degradation of various hydrocarbons in the main body of the manuscript (M&M). They are presented in SI, however I didn’t see them in the M&M. This will make it easier for the reader to follow the manuscript.
- Reference 2 – it seems that first letter of the author’s name is missing
In my opinion, the manuscript is worth publishing in Microorganisms. Therefore, I would recommend minor revision.
Author Response
(1) Q: “...manuscript (especially in the result section) is a bit too detailed and too long.”
A: We reduced the length of Results section text (particularly 3.3 and 3.4). We hesitate to extent discussion more in the case of possible new hydrocarbon degradation related taxa.
(2) Q: “The explanation of one of the most interesting aspects of this manuscript, namely that the gene counts indicated that the actual potential to hydrocarbon degradation was the highest in clean sea ice and lowest in SIO metagenome was a bit insufficiently discussed. I would expect to obtain opposite results, thus the proper discussion of this issue is missing.”
A: The reason for this is following. Current study is based on only shotgun metagenome data with limited sequencing depth. In order to eloborate more on possible new HDO taxa or HDO taxa with with more diverse metabolic capablities for oil degradation, we need multi-omic data set and higher sequencing depth. Based on obtained results of this manuscript we performed a new lab experiement with more treatments and replicates, temporal sampling and combination of different molecular microbiology methods. We hope that the results of this new experiment will provide solid proof for findings of this study. However, we have discussed it in section 4.2, page 19.
(3) Q: What is the authors' opinion on the incubation of ice for a longer period of 6-12 months?
A: Our next project with water samples from same area will answer that question more precisely.
(4) Q: “Please indicate precisely what does it mean “shortly”.”
A: Thank you, we have rephrased it and it means “within 24 hours”.
(5) Q: “Figure 1: I would recommend showing data not vertically but horizontally.”
A: We prefere not to change the figure since some of the graphs are already presented vertically.
(6) Q: ““A total of 350 genera out of 368 genera” please indicate the figure/table where the Reads can verify this information.”
A: 368 genera concidered as hydrocarbon degrading organisms (HDOs) can be found from article Nõlvak et. al. (2021) https://doi.org/10.3390/microorganisms9122425 which is also described in the Methods section 2.4.1. And some additional genera also described as HDOs can be found from Table S2 in this manuscript.
(7) Q: “Figure 3: Please add red dashed lines also in zoomed windows.”
A: Thank you for suggestion and we have added red dashes.
(8) Q: “Is MAH abbreviation explained in the manuscript?”
A: Abbreviation of MAH is explained in M&M section 2.4.3.
(9) Q: “It would be good to indicate genes related to the degradation of various hydrocarbons in the main body of the manuscript (M&M).”
A: We have considered the option before and decided to have the table in SI mainly because as you have stated before as well the manuscript is already quite long and additional table in M&M section would make the manuscript even longer. For reades convenience we have indicated functions for genes in main text, for example in Results section 3.4.1: “long-chain alkane degradation-related almA”.
(10) Q: “Reference 2 – it seems that first letter of the author’s name is missing.”
A: Thank you for noticing, the letter has been added.